# Exosomes Derived from Induced and Wharton’s Jelly-Derived Mesenchymal Stem Cells Promote Senescence-like Features and Migration in Cancer Cells

**DOI:** 10.3390/ijms26136178

**Published:** 2025-06-26

**Authors:** Nidaa A. Ababneh, Razan AlDiqs, Sura Nashwan, Mohammad A. Ismail, Raghda Barham, Renata M. Alatoom, Fairouz Nairat, Mohammad H. Gharandouq, Talal Al-Qaisi, Abdalla Awidi, Tareq Saleh

**Affiliations:** 1Cell Therapy Center, The University of Jordan, Amman 11942, Jordan; razan.nezar.97@gmail.com (R.A.); nashwansura@gmail.com (S.N.); mohd.ismail2@yahoo.com (M.A.I.); raghda.barham@gmail.com (R.B.); alatoomrenata@gmail.com (R.M.A.); fairouzomarnairat@gmail.com (F.N.); mohd.gharandouq@outlook.com (M.H.G.); abdalla.awidi@gmail.com (A.A.); 2South Australian ImmunoGENomics Cancer Institute, Adelaide Medical School, University of Adelaide, Adelaide, SA 5000, Australia; 3Department of Biomedical Sciences, College of Health Sciences, Abu Dhabi University, Abu Dhabi P.O. Box 59911, United Arab Emirates; talal.alqaisi@adu.ac.ae; 4Department of Medical Laboratory Sciences, Faculty of Allied Medical Sciences, Al-Ahliyya Amman University, Amman 19328, Jordan; 5Department of Hematology and Oncology, Jordan University Hospital, Amman 11942, Jordan; 6Hemostasis and Thrombosis Laboratory, School of Medicine, The University of Jordan, Amman 11942, Jordan; 7Department of Pharmacology & Therapeutics, College of Medicine & Health Sciences, Arabian Gulf University, Manama P.O. Box 26671, Bahrain; 8Department of Pharmacology and Public Health, Faculty of Medicine, The Hashemite University, Zarqa 13133, Jordan

**Keywords:** Wharton’s jelly-derived mesenchymal stem cells (WJMSCs), cell-free therapy, exosomes, induced mesenchymal stem cells (iMSCs), senescence, cancer

## Abstract

Mesenchymal stem cell-derived exosomes (MSC-Exos) play a key role in tissue repair, immune regulation, and cancer biology. Due to limitations in MSC expansion and source variability, interest has shifted to induced pluripotent stem cell-derived MSCs (iMSCs) as a promising alternative. This study compares effects of exosomes derived from iMSCs (iMSC-Exos) and Wharton’s jelly MSCs (WJMSC-Exos) on MCF7 and A549 cancer cells. Both types of exosomes reduced MCF7 proliferation and induced a senescence-like state, rather than apoptosis, although the antiproliferative effect was transient in A549 cells. Notably, WJMSC-Exos promoted migration in both MCF7 and A549, whereas iMSC-Exos did not exhibit this effect. Overall, WJMSC-Exos had a more robust impact on cancer cell proliferation and migration. These findings highlight the diverse effects of exosomes on cancer and the development of a senescence-like state as an important response to Exos exposure. Moreover, these findings invite for more careful evaluation of the therapeutic role of iMSC-derived Exos.

## 1. Introduction

Stem cells are undifferentiated cells that have the capacity of self-renewal and giving rise to specialized cells under specific conditions [1]. Various types of stem cells can be obtained from different sources with different differentiation potentials [2,3]. Mesenchymal stem cells (MSCs) are highly valued in cancer therapy and regenerative medicine due to their remarkable ability to migrate to tissues affected by malignancy or tissue damage [4]. MSCs are adult, non-hematopoietic stem cells found in various tissues, including bone marrow, adipose tissue, bone, Wharton’s jelly, umbilical cord blood, and peripheral blood [1]. Based on the International Society for Cellular Therapy (ISCT) guidelines, a MSC must possess the following: (a) adherence to plastic under appropriate culture conditions; (b) expression of surface markers CD29, CD44, CD73, CD90, and CD105, and lack of hematopoietic (CD45 and CD14), endothelial (CD31 and CD34), and HLA-DR markers; and (c) the ability to differentiate into adipocytes, osteoblasts, and chondrocytes in vitro [5]. MSCs are recognized for their considerable potential in regenerative medicine. However, challenges such as inconsistent supply, varying quality, and changes in phenotype due to prolonged expansion have impeded their effective use [6].

Induced pluripotent stem cells (iPSCs) are generated by reprogramming adult cells, giving them similar differentiation potential as embryonic stem cells (ESCs) [7], thus, opening wide avenues for research and therapy especially considering their immunological advantage in autologous transplantation [8]. iPSCs can differentiate into MSCs, offering great potential by overcoming issues of heterogeneity, complex mechanisms of action, and the demand in production processes associated with conventional MSCs [9]. Thus, induced pluripotent stem cell-derived mesenchymal stem cells (iMSCs) provide a promising alternative source of MSCs combining both advantages of iPSCs and traditional MSCs [10].

While some advantages of MSC therapy may arise from ‘trans-differentiation’, the majority of MSC effects are attributed to paracrine mechanisms [11]. In paracrine signaling between MSCs and neighboring cells, MSCs produce and release specific substances, including trophic factors, cytokines, chemokines, hormones, and extracellular vesicles (EVs) like exosomes (Exos), which are secreted to facilitate intercellular communication [12]. Growing evidence indicates that MSC-derived extracellular vesicles (MSC-EVs) play a crucial role in stimulating tissue regeneration, repair, and immune regulation [13]. Moreover, using conditioned medium or EVs in cell-free therapy offers several advantages over cell-based therapy, as it mitigates the risks associated with stem cell use and allows for easier storage and greater efficiency in therapeutic applications [14]. MSC-derived exosomes (MSC-Exos) are natural, nano-sized particles enclosed by a lipid bilayer, designed to deliver cargos from MSCs—such as nucleic acids, lipids, and proteins—into the target cell [8]. Exos are becoming increasingly popular among EV subpopulations due to their significant role in immune modulation and regenerative therapy [14,15,16].

Although exosomes derived from primary MSC sources, such as WJMSC, have demonstrated therapeutic potential, iMSCs emerge as promising, scalable, and ethically favorable alternative. WJMSC are commonly used in MSC research due to their high expansion capacity, immunomodulatory properties, and ease of isolation without apparent ethical concerns [17]. However, the comparative biological effects of iMSC-derived exosomes (iMSC-Exos) and WJMSC-derived exosomes (WJMSC-Exos), particularly on cancer cells, remain insufficiently explored. This study, therefore, aims to evaluate the potential of iMSC-Exos by comparing their effects on breast and lung cancer cell models (MCF7 and A549, respectively). The comparison conducted in this work focuses on their ability to affect tumor cell proliferation, senescence, and migration. By exploring these effects, this study aims to evaluate the potential of iMSC-Exos for cell-free cancer therapy and to explore their potential as a reliable alternative to conventional MSC-Exos.

## 2. Results

### 2.1. Characterization of WJMSCs and iMSCs, Their Derived Exosomes, and Uptake by Fibrobalsts

To characterize iMSCs and WJMSCs, flow cytometry was utilized to analyze the presence of human mesenchymal stem cell (hMSC) surface markers, specifically CD90, CD105, CD73, and CD44, while also assessing the absence of negative hMSC markers, including CD34, CD45, CD14, CD11b, CD79a, CD19, and HLA-DR. The results indicated successful expression of the positive hMSC markers and the absence of negative markers in both iMSCs and WJMSCs, with no significant differences observed, confirming effective differentiation of iPSCs into iMSCs (Appendix A). Furthermore, osteogenic and adipogenic differentiation assays were performed to evaluate the differentiation capabilities of iMSCs and WJMSCs into osteogenic and adipogenic lineages, respectively (Appendix A). Both cell types exhibited successful differentiation potential in these assays.

The detection of exosome surface markers CD9, CD81, and CD63 was performed via flow cytometry using sulfate-latex beads to bind the Exos, as outlined in the methods section. The results demonstrated successful expression of these markers in both WJMSC-Exos and iMSC-Exos (Appendix A). Notably, both types of Exos exhibited different levels of surface marker expression, with a non-significantly higher expression of the CD9 tetraspanin protein in both iMSC-Exos and WJMSC-Exos compared to CD81 and CD63 Appendix A). However, no significant differences in the expression of exosome surface markers were observed between the two groups. The average percentages of surface markers in WJMSC-Exos were as follows: CD9 (86%), CD81 (75%), and CD63 (50%). For iMSC-Exos, the average percentages were CD9 (89%), CD81 (73%), and CD63 (46%) (Appendix A). Transmission electron microscopy (TEM) analysis of both iMSC-Exos and WJMSC-Exos confirmed a characteristic cup-like morphology (Appendix A).

Size distribution measurements revealed a heterogeneous size profile for iMSC-Exos, with a size range of 59 nm, while WJMSC-Exos exhibited a more homogeneous size distribution, averaging 104 nm (Appendix A). Dynamic light scattering analyses further indicated the presence of an additional population of iMSC-Exos with sizes ranging from 460 nm to 5 μm (Appendix A). To evaluate the uptake of Exos in fibroblast (control), cells were treated with Exos derived from iMSCs and WJMSCs. Exos were labeled with the fluorescent dye DiI-O, which integrates into lipid bilayer membranes and emits an orange-red fluorescence upon their uptake. Cells were also labeled with CMFDA (green) to highlight the cell body and DAPI (blue) to stain the nucleus. Following 12 h of incubation, cells were fixed and observed using fluorescent microscopy, images revealed red fluorescent particles in the perinuclear regions of fibroblast cells, indicating successful internalization of the Exos into fibroblast cells (Appendix A).

### 2.2. iMSC- and WJMSC-Derived Exosomes Suppress Proliferation in MCF7 and A549 but with Minimal Apoptosis

To assess the uptake of Exos by MCF7 and A549 cancer cells, these cells were treated with DiI-O labeled Exos, following the same protocol used for HDFs. CMFDA staining (green) was employed to visualize the cell bodies, while DAPI (blue) was used to stain the nuclei. After a 12-h incubation period, the cells were fixed and examined using fluorescence microscopy (Figure 1A,B). Immunofluorescent images demonstrated red fluorescent particles localized in the perinuclear regions of MCF7 and A549 cells, indicating successful internalization of the Exos in both cancer cell types (Figure 1A,B).

To assess cell proliferation, cancer cell lines MCF7 and A549, along with fibroblasts (HDF), were treated with 50 μg/mL of either iMSC- or WJMSC-Exos for 24 and 48 h. Serum-free medium (SFM) was used as the negative control, as it lacks serum components, including fetal bovine serum (FBS), which is known to contain EVs such as Exos [18]. Cell viability was evaluated using the MTT assay. The results indicated that MCF7 cell proliferation decreased following treatment with both types of Exos at both time points, with no significant differences between the two Exos types, highlighting their potential inhibitory effects on cancer cells (Figure 1C). In contrast, A549 cells exhibited reduced proliferation at 24 h after treatment with either Exos type; however, this effect was not significant at 48 h, suggesting a transient antiproliferative effect of both iMSC- and WJMSC-derived Exos on A549 cells (Figure 1D). Additionally, fibroblast cells were used as a control and displayed no significant changes in proliferation (Figure 1E), indicating that the antiproliferative effects of Exos may be cancer cell-selective.

Flow cytometric analysis via Annexin-V/PI staining was performed to assess the ability of iMSC and WJMSC-Exos to induce apoptosis in MCF7 and A549 cells after 48 h of treatment. In MCF7 cells, the percentages of early apoptotic (Annexin V+/PI−), late apoptotic (Annexin V+/PI+) and necrotic (Annexin V−/PI+) cells showed no statistically significant differences in response to treatment with both Exos types compared to SFM negative control (Figure 1F). Similarly, A549 cells showed no significant effect on apoptosis after treatment with either type of Exos (Figure 1G).

### 2.3. iMSC- and WJMSC-Derived Exos Induce a Senescence-like State in MCF7 and A549 Cells

Given that iMSC- and WJMSC-derived Exos induced minimal apoptosis in both cancer cell lines, we explored another cell stress response to elucidate their antiproliferative effects. Senescence, a well-established cell stress response, is characterized by a persistent growth arrest along with distinct transcriptomic and secretory profiles [19]. Additionally, recent studies have established a link between senescence and Exos [20]. In our model, after 48 h of treatment with either type of Exos or SFM negative control (untreated), cancer cells were stained for SA-β-Gal to assess the potential induction of a senescence-like response. The percentage of cells positive for SA-β-Gal activity was determined by counting the number of blue/green-stained cells within the total population [21]. Results indicated a significantly higher percentage of SA-β-Gal-positive cells in MCF7 when treated with either type of Exos compared to the control (Figure 2A,B). Similarly, treatment of A549 cells with either type of Exos led to a statistically significant increase in the percentage of SA-β-Gal-positive cells, as observed microscopically (Figure 2C,D). Moreover, since the investigation of a senescence-like phenotype requires testing additional senescence-related markers [22], we measured the gene expression of p53 and p21^Cip1^, as well as several senescence-associated secretory phenotype (SASP)-related components, namely, CXCL8, IL-6, TNF-α and TGF-β1. Our data shows that in MCF7 cells, both Exos types induced upregulation of p53 (Figure 2E) and p21^Cip1^ (Figure 2F), although the increase in p21^Cip1^ was not statistically significant. In A549 cells, p53 expression was significantly elevated only following iMSC-Exos treatment (Figure 2E), whereas p21^Cip1^ was upregulated by both exosome types (Figure 2F). A significant upregulation of CXCL8 was observed in response to both iMSC- and WJMSC-derived Exos in both cancer cell lines (Figure 2G). IL-6 expression was also increased under all treatment conditions; however, the increase was not statistically significant in MCF7 cells treated with WJMSC-Exos (Figure 2H). Similarly, both TNF-α and TGF-β1 were upregulated in response to iMSC- and WJMSC-Exos in both MCF7 and A549 cells (Figure 2I and Figure 2J, respectively). Collectively, these findings suggest that Exos treatment could favor a senescence-like state rather than apoptosis in both studied cancer cell lines.

### 2.4. Enhanced Migratory Potential in MCF7 and A549 Cells in Response to WJMSC-Exos but Not iMSC-Exos

A scratch assay was performed on MCF7 and A549 cancer cells to measure the rate of wound healing at approximately 0, 9, 20, and 47 h after the scratch was made. Microscopic images of MCF7 (Figure 3A) and A549 (Figure 3C) cells were captured to assess migration. In MCF7 cells, treatment with WJMSC- Exos resulted in a significant increase in cell migration compared to both the SFM control and iMSC-Exos, with no statistically significant effect observed with iMSC-Exos (Figure 3B). In A549 cells, a similar effect of WJMSC-Exos was observed, with variations at specific time point of wound healing measurement (Figure 3D). WJMSC-Exos significantly increased migration at ~9 h compared to iMSC-Exos, and at 47 h compared to both SFM and iMSC-Exos, with the most pronounced effect seen at approximately 47 h.

## 3. Discussion

Recently, iMSCs have emerged as a promising alternative to traditional MSCs, addressing limitations such as limited expansion potential and source heterogeneity [23,24]. However, further research is needed to thoroughly compare the properties of iMSCs and MSCs, including potential differences in the characteristics and functional effects of Exos derived from these sources. While direct comparative studies remain limited, some have reported distinct molecular and functional differences between iMSC- and MSC-derived Exos [25,26,27,28]. Additionally, other investigations have highlighted the regenerative and immunomodulatory potential of iMSC-derived Exos in various disease models [29,30,31,32], highlighting the importance of considering their unique features when evaluating biological outcomes. Although Exos hold promises as therapeutic agents for cancer treatment, further research is required to thoroughly understand their mechanisms of action and potential side effects before they can be safely utilized in clinical practice [33,34]. Therefore, this study examines the effects of Exos derived from iMSCs and WJMSCs on breast and lung cancer cells (represented by MCF7 and A549 cell lines, respectively). Our findings aim to contribute to the ongoing efforts to explore and understand the relationship between cancer and MSCs, shedding light on the potential therapeutic applications of iMSC- versus MSC-derived Exos in cancer treatment and regenerative medicine. Both iMSCs and WJMSCs utilized in our study fulfilled the criteria set by the ISCT for identifying MSCs, exhibiting key stem cell characteristics including plastic adherence, expression of specific MSC surface markers (CD90, CD105, CD73, and CD44), and the ability to differentiate into osteogenic and adipogenic lineages [35].

Exos isolated from the conditioned medium of iMSCs and WJMSCs were characterized by the presence of Exos surface markers, including CD9, CD81, and CD63 [36]. The expression of these surface markers was not significantly different between the two groups, with both iMSC- and WJMSC-derived Exos showing high expression of the CD9 tetraspanin protein. TEM revealed aggregates of Exos in iMSC-Exo samples, likely due to light scattering from nanoparticle clusters being detected as a single larger particle. These aggregates may indicate that the Exos concentration in iMSC-Exo samples is high. Moreover, the nanoparticle size distribution of isolated Exos from both groups was evaluated using DLS and showed a typical size range of Exos, with greater size heterogeneity and large size measurements observed in iMSC-Exos [37,38,39]. In addition, morphology analysis using TEM revealed that both iMSCs- and WJMSCs-Exos exhibited a similar morphological structure, consistent with the cup-like shaped vesicles [40]. Overall, the isolated Exos exhibited the typical characteristics associated with Exos, validating the efficacy of the isolation methods implemented in this study.

This study demonstrated a significant reduction in the viability of MCF7 and A549 cells when treated with either iMSC- or WJMSC-Exos, suggesting a potential antiproliferative effect. Importantly, this antiproliferative effect was more predominant in cancer cells compared to non-malignant fibroblasts suggesting tumor cell-selectivity. It is noteworthy that, in A549 cells, an early antiproliferative effect was observed at 24 h post-treatment with either Exos type, but this effect diminished and became statistically insignificant after 48 h. Surprisingly, although iMSC- and WJMSC-Exos resulted in reduced proliferation in both MCF7 and A549 cells, apoptosis levels were minimal. Alternatively, our analysis showed an increase in senescent-like cell density marked by SA-β Gal upregulation following treatment with either type of Exos. Cellular senescence represents a gradual loss of proliferative capacity [41] and serves as a potent barrier against tumorigenesis [42]. Therefore, the antiproliferative potential of these Exos might be mediated through the induction of a senescence-like state in both MCF7 and A549 cells. Furthermore, the expression of senescence-associated genes, including p53, p21^Cip1^, IL-6, CXCL8, TNF-α, and TGF-β1, was upregulated following Exos treatment, consistent with the SA-β-Gal staining results. While MSC-Exos were found to reduce the likelihood for senescence induction [43,44,45], recent studies have highlighted a role for Exos as key modulator of cellular senescence and a main component of SASP [46]. Importantly, senescent cells have been shown to amplify the senescence state through the release of Exos in a non-cell-autonomous fashion [47,48,49]. This highlights the influence of cell types and conditions on the observed senescence-like state, whether it increases or decreases. Overall, these findings suggest that the Exos derived from iMSCs and WJMSCs might favor a senescence-like state in cancer cells.

Additionally, the measurement of migration rates in MCF7 and A549 cells treated with WJMSC-Exos showed an increased migration rate, with a more consistent and statistically significant effect in MCF7 cells. In contrast, migration in A549 cells was not significant at approximately 9 h when assessing wound healing. This effect strikingly contrasts with the antitumor effects observed in the proliferation assays in both MCF7 and A549 cells. More importantly, this finding contradicts the widely accepted migration/proliferation hypothesis, which typically suggests a positive correlation between migration and proliferation [50,51]. However, a study by Garay et al. examined the relationship between migration and proliferation across various cancers, including several lung cancer cell lines (though A549 was not included), and found variability in results [52]. They proposed that tumors originating from different organs might regulate migration and proliferation in distinct ways [52]. Additionally, previous studies have reported results consistent with our findings, where migration and proliferation were negatively correlated [53,54]. Moreover, senescence has been strongly linked to increased invasiveness of tumor cell population. Several studies have shown that increased senescence correlates with decreased proliferation but enhanced migration [55,56,57,58]. In this context, the SASP plays a crucial role in promoting tumor-related processes such as angiogenesis, stem cell-like behavior, genotoxicity, chronic inflammation, invasion, migration, and immunosuppression [59]. In fact, SASP in cancer is recognized as a double-edged sword: while it can help eliminate cancer cells through immune activation, its persistent signaling has also been associated with tumorigenesis and epithelial-mesenchymal transition [59,60]. These findings suggest that a senescence-like state may help explain the decreased proliferation observed in our cancer cells, as well as their elevated migratory potential (Appendix A). However, further investigation is necessary to elucidate the underlying factors governing these behaviors. These findings also indicate that despite the senescence-mediated antiproliferative effect of Exos, the induction of a senescence-like state in cancer cells might not be a favorable outcome.

In comparison with previous studies, our results add to the ongoing debate regarding the dual nature of MSC-derived Exos in cancer biology. Evidence indicates that MSC-Exos possess both pro-tumorigenic and anti-tumorigenic properties [13,61,62]. For example, while some studies demonstrate tumor-suppressive effects—primarily mediated through the delivery of miRNAs from Exos derived from different MSC sources across various cancer types [63,64,65,66], others report tumor-promoting effects, linked to pro-angiogenic and pro-metastatic cargo, including certain miRNAs [67,68,69]. These discrepancies largely reflect the heterogeneity of MSC-Exos that stems from MSC source variability, culture conditions, and Exos isolation methods, all of which significantly alter cargo composition, including distinct miRNA, cytokine, and protein profiles [70]. The tumor model selected for study also plays a critical role in shaping the observed effects [71,72]. For example, Exos derived from adipose tissue MSCs have been shown to enhance proliferation and migration of MCF7 breast cancer cells in vitro [73,74], whereas the same Exos inhibited proliferation and impaired wound repair in ovarian cancer cell models such as SKOV-3 and A2780 [63].

Overall, iMSC-Exos exhibited comparable effects to WJMSC-Exos in reducing proliferation and inducing a senescence-like state in cancer cell models. However, only WJMSC-Exos promoted migration in both models, with varying degrees of effects observed between cell types. These findings support the potential of iMSCs as a scalable and consistent alternative to WJMSCs, while also highlighting source-dependent differences in functional properties, particularly regarding cancer cell migration. Despite the insights gained, this study is limited by the use of only one cell line per cancer type (MCF7 and A549) and the absence of in vivo testing, which may not capture the full heterogeneity of the observed responses. Another key limitation is the lack of senescence examination in non-cancerous cell models, leaving it unclear whether the induction of a senescence-like state by iMSC- and WJMSC-Exos is specific to cancer cells. Additionally, the mechanisms underlying the effects on proliferation, senescence, and migration require further investigation, particularly with a focus on the exosomal cargo responsible for these changes. Optimization of experimental conditions, such as Exos dosage and exposure time, was also limited and should be addressed in future work. Finally, a thorough comparison between the ability of MSC- and iMSC-derived Exos to induce senescence-like features should be conducted prior to the consideration of iMSC as a therapeutic replacement. Future studies should also incorporate dose-response experiments and ELISA analyses of conditioned media to evaluate SASP factors, along with comprehensive senescence-specific transcriptomic profiling, to better elucidate the role of senescence in mediating the observed effects of iMSC-Exos on tumors cells.

## 4. Materials and Methods

### 4.1. Cell Culture

The Human Wharton’s jelly-derived mesenchymal stem cells (WJMSCs) were previously prepared at the Cell Therapy Center (University of Jordan, Amman, Jordan) [75]. This study was approved by the Institutional Review Board (IRB) No: IRB/2021/4 at the Cell Therapy Center/University of Jordan. iMSCs and WJMSCs at passage 3 were cultured in cell culture medium (CCM) consisted of Essential Medium Eagle-Alpha Modification (α-MEM, Gibco, Thermo Fisher Scientific, Waltham, MA, USA) supplemented with 15% Fetal Bovine Serum (FBS, HyClone, Cytiva, Logan, UT, USA), 1% of 100× Glutamax (Gibco, Thermo Fisher Scientific, Waltham, MA, USA), and 1% of 100× antibiotic-antimycotic mixture (Gibco, Thermo Fisher Scientific, Waltham, MA, USA). MCF7 (breast cancer, ATCC^®^ HTB-22^™^, Manassas, VA, USA) and A549 (alveolar basal epithelial adenocarcinoma, ATCC^®^ CCL-185^™^, Manassas, VA, USA) cancer cell lines were cultured in RPMI (Gibco, Thermo Fisher Scientific, Waltham, MA, USA) supplemented with 10% FBS, 1% of 100× Glutamax (Gibco, Thermo Fisher Scientific, Waltham, MA, USA), and 1% of 100× antibiotic antimycotic (Gibco, Thermo Fisher Scientific, Waltham, MA, USA). Cell culture media were exchanged every two days, and all cells were incubated in standard culture conditions (37 °C, 21% O_2_, and 5% CO_2_).

### 4.2. Generation of iMSCs

In this study, four iPSC lines previously derived from dermal fibroblasts [76,77] were utilized. The iPSCs were cultured on Matrigel-coated plates (Corning, Corning, NY, USA) and maintained in mTeSR medium. To detach the monolayer cultures, 0.5 M EDTA (Thermo Fisher Scientific, Waltham, MA, USA) was used, after which the cell suspensions were cultured in ultra-low attachment plates (Thermo Fisher Scientific, Waltham, MA, USA) to form embryoid bodies (EBs) in MSC differentiation media composed of Alpha MEM (Gibco, Thermo Fisher Scientific, Waltham, MA, USA) supplemented with 15% FBS (FBS, HyClone, Cytiva, Logan, UT, USA), 1% Glutamax (Gibco, Thermo Fisher Scientific, Waltham, MA, USA), and 1% antibiotic-antimycotic mixture (Gibco, Thermo Fisher Scientific, Waltham, MA, USA). The medium was refreshed on days 2 and 4 of differentiation, with 10 μM and 0.1 μM retinoic acid (RA, Sigma-Aldrich, St. Louis, MO, USA) added respectively. On day 6, the media was switched to a RA-free differentiation medium. For the generation of EB-iMSCs, the EBs were transferred to Matrigel-coated plates on day 7 and cultured in the above mentioned MSC differentiation media and medium was was exchanged every two days. Starting on day 12, the medium was supplemented with 2.5 ng/mL basic fibroblast growth factor (bFGF, PeproTech, Cranbury, NJ, USA) and exchanged every two days. Once the iMSCs reached 80–90% confluency, they were passaged and cryopreserved in FBS containing 10% DMSO, then stored in liquid nitrogen [78,79].

### 4.3. Differentiation of iMSCs and WJMSCs

Osteogenic Differentiation: Cells were seeded in triplicate in 6-well tissue culture plates (SPL Life Sciences Co., Ltd., Pocheon-si, Gyeonggi-do, Republic of Korea) at a density of 200,000 cells per well and cultured in CCM until reaching at least 50% confluency. At that point, the medium was replaced with osteogenic differentiation medium, composed of α-MEM supplemented with 15% FBS, 1% 100× Glutamax, 1% 100× antibiotic-antimycotic solution, 10 mM Dexamethazone (Sigma-Aldrich, St. Louis, MO, USA), 50 μg/mL ascorbic acid 2-phosphate (Sigma-Aldrich, St. Louis, MO, USA) and 10 mM β-glycerophosphate (Carbosynth, Compton, UK). The cells were cultured in this differentiation medium for 21–28 days or until calcium deposits were visible. Control cells remained in CCM, with media changes every 2–3 days. Upon detection of mineral deposits, one well from each sample was stained with 2% Alizarin Red in distilled water (Sigma-Aldrich, St. Louis, MO, USA) to visualize calcium deposition. The stained deposits were then examined and imaged using the EVOS XL Core Imaging System (Thermo Fisher Scientific, Waltham, MA, USA).

Adipogenic Differentiation: iMSCs and ADMSCs were seeded in triplicate in 6-well tissue culture plates (SPL Life Sciences Co., Ltd., Pocheon-si, Gyeonggi-do, Republic of Korea) at a density of 200,000 cells per well and cultured in complete culture medium (CCM) until they reached at least 50% confluency. At this point, the medium was replaced with adipogenic differentiation medium, composed of Minimum Essential Medium Eagle-Alpha Modification (Alpha MEM, Gibco, Thermo Fisher Scientific, Waltham, MA, USA) supplemented with 15% fetal bovine serum (FBS, Hyclone), 1% 100× Glutamax (Gibco, USA), 1% 100× antibiotic-antimycotic solution (Gibco, Thermo Fisher Scientific, Waltham, MA, USA), 10 mM dexamethasone (Sigma-Aldrich, St. Louis, MO, USA), 500 μM 3-isobutyl-1-methylxanthine (IBMX) (Sigma-Aldrich, St. Louis, MO, USA), 0.2 mM indomethacin (Sigma-Aldrich, St. Louis, MO, USA), and 10 μg/mL insulin (Sigma-Aldrich, St. Louis, MO, USA). The cells were cultured in this medium for 14–21 days, or until fat vacuoles became visible.

Once adipogenic differentiation was confirmed, the cells were stained with 0.3% Oil Red-O in isopropanol (both from Sigma-Aldrich, St. Louis, MO, USA) to visualize the fat vacuoles, which were then examined and photographed using the EVOS XL Core Imaging System (Thermo Fisher Scientific, Waltham, MA, USA).

### 4.4. Characterization of iMSCs and WJMSCs

For the assessment of hMSC surface markers in iMSCs and WJMSCs, the expression of CD90, CD105, CD73, and CD44 was evaluated using the Human MSC Analysis Kit (BD Biosciences, San Jose, CA, USA) according to the manufacturer’s instructions. The absence of Nanog and Tra-1-60 was assessed as described in our previous publication [76].

Cells were harvested at early passages (passage < 8) using trypsin (Gibco, Thermo Fisher Scientific, Waltham, MA, USA), washed with 1× PBS (Gibco, Thermo Fisher Scientific, Waltham, MA, USA), and resuspended in 800 μL of 1% bovine serum albumin (BSA, Gibco, Thermo Fisher Scientific, Waltham, MA, USA) staining buffer. The cells were then incubated in the dark for 30 min with antibodies conjugated to CD90-FITC, CD105-perCP, CD73-APC, CD73-PE, and CD44-PE, or corresponding isotype control antibodies (BD Biosciences, San Jose, CA, USA). Antibody concentrations were adjusted to 50 μg/mL for CD90-FITC, CD105-PerCP, and CD73-APC, while CD44-PE and the isotype control antibodies were used at 20 μg/mL. After incubation, the cells were washed twice with 1× PBS and resuspended in 200 μL of PBS. Surface marker detection was performed using a BD FACS Canto II flow cytometer (BD Biosciences, Franklin Lakes, NJ, USA), and the results were analyzed using BD FACSDiva software v8.0.2 (BD Biosciences, Franklin Lakes, NJ, USA).

### 4.5. Purification of iMSC-Exos and WJMSC-Exos

iMSCs and WJMSCs were seeded in T175 at passage 3 and maintained in cell culture medium until confluency and then medium was aspirated and cells were washed two times with 1× PBS, then serum free medium was used and kept for 48 h. Then conditioned medium from iMSCs and WJMSCs was collected and centrifuged at 300× *g* for 10 min at 4 °C. The supernatant was then collected and centrifuged again at 2000× *g* for 20 min at 4 °C to remove any remaining cells and large apoptotic bodies. The resulting supernatant was filtered through a 0.22 μm filter (TPP Techno Plastic Products AG, Trasadingen, Switzerland) for further exosome purification. After filtration, the filtrate was ultracentrifuged at 110,000× *g* for 2 h at 4 °C to pellet the Exos. The exosome pellet was washed with a large volume of filtered phosphate-buffered saline (PBS; Gibco, Thermo Fisher Scientific, Waltham, MA, USA) and ultracentrifuged again at the same speed for another 2 h. Finally, the Exos were resuspended in 500 μL of filtered PBS [80], and their concentration was measured using a Micro BCA Protein Assay kit (Abcam, Cambridge, UK). Exos were then diluted to a final concentration of 100 µg/mL for downstream applications.

A total of ~100 μg of the isolated Exos was incubated with 3 μL aldehyde/sulfate-latex beads (4% *w*/*v*, 4 μm; Invitrogen, Thermo Fisher Scientific, Waltham, MA, USA) for 15 min at room temperature. Afterward, 1× filtered PBS was added to the Exo-bead mixture, which was then incubated overnight at 4 °C with gentle shaking. To block non-specific binding, the Exo-beads were incubated with 1M glycine (Sigma-Aldrich, St. Louis, MO, USA) for 30 min at room temperature. The Exo-beads were then incubated with specific antibodies, including anti-CD9 (A5488), anti-CD81 (AF647), and anti-CD63 (APC) (all from BD Biosciences, Franklin Lakes, NJ, USA), along with their respective isotype controls. Following antibody incubation, 1× filtered PBS was added to the Exo-beads prior to the characterization of exosome surface markers using a BD FACS Canto II, and the results were analyzed with BD FACSDiva software.

### 4.6. Size Distribution of Exosomes

The size distribution of ~100 μg of Exos was evaluated using dynamic light scattering (DLS, Malvern Panalytical, Malvern, Worcestershire, UK), which utilizes light scattering and Brownian motion properties. Exos were resuspended in 1 mL of PBS (Gibco, Thermo Fisher Scientific, Waltham, MA, USA), mixed thoroughly, and injected into the ZetaView instrument (Malvern Nano ZS, Worcestershire, UK). Measurements were performed according to the manufacturer’s instructions for exosome analysis, using the following parameters: 173° backscatter measurement angle, 10 runs per sample, 60-s run duration, 3 measurements per sample, with a 10-s delay between each measurement. Data were analyzed using Zetasizer software version 7.11 (Malvern Panalytical, Malvern, Worcestershire, UK).

### 4.7. Exosome Morphology

To examine the morphology of Exos, ~100 μg of purified Exos were resuspended in a 1:1 mixture with 2% PFA (Sigma-Aldrich, St. Louis, MO, USA) and deposited onto Formvar-carbon-coated electron microscopy (EM) grids. This allowed the Exos to adsorb onto the membrane for 20 min in a dry environment at room temperature. The grids were then washed and immersed in drops of 1% glutaraldehyde (Sigma-Aldrich, St. Louis, MO, USA) for 5 min. Following this, the grids were washed seven times with distilled water, with each wash lasting 2 min. To enhance contrast, the grids were placed on drops of uranyl-oxalate (Electron Microscopy Sciences, Hatfield, PA, USA) for 5 min and then transferred to methyl cellulose-UA (Electron Microscopy Sciences, Hatfield, PA, USA) for 10 min on ice. After air-drying for 10 min, the grids were examined using a Versa 3D FEI transmission electron microscope (TEM, Thermo Fisher Scientific, Hillsboro, OR, USA) at an acceleration voltage of 30 V.

### 4.8. Cellular Uptake of Exosomes

The isolated iMSC-Exos and WJMSC-Exos at a final concentration of 100 µg/mL were labeled with DiI fluorescent dye according to the manufacturer’s instructions (Invitrogen, Thermo Fisher Scientific, Waltham, MA, USA). In brief,10 µM DiI fluorescent dye was added to the isolated Exos, and the mixture was incubated for 1 h at room temperature. To eliminate excess DiI dye, the labeled Exos were concentrated further by ultracentrifugation at 110,000× *g* for 1.5 h at 4 °C, followed by re-suspension in filtered PBS.

MCF7 or A549 cells (1.5 × 10^5^ cells/well) were seeded in serum-free RPMI (Invitrogen, Thermo Fisher Scientific, Waltham, MA, USA) and treated with 50 μg/mL of the DiI-labeled Exos for 16 h. After treatment, the cells were washed with PBS and incubated with CMFDA dye (Invitrogen, Thermo Fisher Scientific, Waltham, MA, USA) for 30 min at 37 °C. The cells were then fixed with 4% paraformaldehyde (PFA) and stained with DAPI (1:1000, Invitrogen, Thermo Fisher Scientific, Waltham, MA, USA). Finally, the cells were mounted using anti-fade mounting medium (Abcam, Cambridge, UK), and images were captured using an inverted fluorescent microscope (Carl Zeiss AG, Oberkochen, Germany).

### 4.9. Cancer Cell Proliferation Assay

MCF-7 and A549 cells were seeded in 96-well plates (SPL Life Sciences Co., Ltd., Pocheon-si, Gyeonggi-do, Republic of Korea) at a density of 8 × 10^3^ cells per well. After 24 h of incubation, the medium was replaced with 100 μL of serum-free RPMI (Invitrogen, Thermo Fisher Scientific, Waltham, MA, USA) containing 50 µg/mL of either iMSC-Exos or WJMSC-Exos. At 24 and 48 h, 10 μL of MTT reagent (Thiazolyl Blue Tetrazolium Bromide; ATCC, Manassas, VA, USA) was added to each well, followed by a 3-h incubation at 37 °C. Subsequently, 100 μL of solubilization stop solution was added, and the plates were incubated for an additional 30 min at 37 °C. Absorbance at 570 nm was then measured using a Biotek Cytation 5, and the data were analyzed with BioTek Gen 5 data analysis software version Gen5 3.10.06 (BioTek Instruments, Winooski, VT, USA).

### 4.10. Apoptosis Assay

MCF7 or A549 cells were seeded at a density of 3 × 10^5^ cells per well in six-well plates (SPL Life Sciences Co., Ltd., Pocheon-si, Gyeonggi-do, Republic of Korea) containing serum-free RPMI (Invitrogen, Thermo Fisher Scientific, Waltham, MA, USA) and treated with 50 µg/mL of either iMSC-Exos or WJMSC-Exos. To assess cell apoptosis, the cells were stained with 5 μL FITC-conjugated Annexin V and 5 μL propidium iodide (PI) (eBioscience™, Thermo Fisher Scientific, Waltham, MA, USA), following the manufacturer’s instructions. The percentage of apoptotic cells was subsequently determined using a BD FACS Canto II flow cytometer (eBioscience™, Thermo Fisher Scientific, Waltham, MA, USA) and analyzed with BD FACSDiva software.

### 4.11. Wound Scratch Assay

Confluent cultures of MCF7 or A549 cells were scratched with a 200 μm tip. Following that, the cells were washed twice with 1× PBS (Gibco, Thermo Fisher Scientific, Waltham, MA, USA), and the medium was replaced with serum-free RPMI (Invitrogen, Thermo Fisher Scientific, Waltham, MA, USA) containing 50 μg/mL of either iMSC-derived iMSC-Exos or WJMSC-Exos. Images of wound closure were captured at approximately 0-, 9-, 20-, and 47-h post-treatment, and the extent of closure was quantified using ImageJ software version 1.54p (NIH, Bethesda, MD, USA).

### 4.12. Senescence Associated β-Galactosidase Staining (SA-βGal) Analysis

Senescence in MCF7 and A549 cells, seeded at a density of 3 × 10⁵ cells per well, was assessed using a senescence detection kit (Abcam, cat #ab65351, Cambridge, UK) according to the manufacturer’s instructions. The cell culture medium was replaced with serum-free RPMI (Invitrogen, Thermo Fisher Scientific, Waltham, MA, USA) containing 50 μg/mL of either iMSC-Exos or WJMSC-Exos, and the cells were incubated for 48 h under standard culture conditions. Following incubation, the cells were washed once with 1× PBS, fixed with a fixative solution, and allowed to incubate for 15 min at room temperature. The cells were then stained with the provided staining solution and incubated overnight at 37 °C. Senescent cells were imaged using a Cytation 5 (BioTek Instruments, Winooski, VT, USA), and the percentage of senescent cells was calculated by counting the number of senescent cells relative to the total cell population [21].

### 4.13. Gene Expression

Total RNA was extracted from cancer cells after exosome treatment using the RNeasy^®^ Mini Kit (Qiagen, Hilden, Germany), following the manufacturer’s instructions. To assess gene expression, quantitative reverse transcription PCR (qRT-PCR) was performed. Briefly, total RNA was reverse transcribed into complementary DNA (cDNA) using the PrimeScript^TM^ RT Master Mix (Takara Bio Inc., Kusatsu, Japan). Each qRT-PCR reactions was carried out in a 20 μL volumes containing 10 ng/μL of cDNA, 0.2 μM gene-specific primer pairs (listed in Appendix A), and the TB Green^®^ Premix Ex Taq^™^ II reagent (Tli RNase H Plus; Takara Bio Inc., Kusatsu, Japan). Reactions were run on a Bio-Rad CFX96 real-time PCR system (Bio-Rad Laboratories, Hercules, CA, USA). Target gene expression levels were normalized to GAPDH and analyzed using the 2^−ΔΔCT^ method.

### 4.14. Statistical Analysis

All data were analyzed using GraphPad Prism version 9.3.1 (GraphPad Software, San Diego, CA, USA). Statistical analyses included one-way analyses of variance (ANOVA) and two-way analyses of variance (ANOVA), followed by the Bonferroni post-hoc test when applicable. A *p*-value of ≤0.05 was considered statistically significant (* ≤ 0.05, ** ≤ 0.01, *** ≤ 0.001, **** ≤ 0.0001). Data are presented as mean ± standard deviation (SD). All experiments were performed in triplicate (*n* = 3).

## 5. Conclusions

In summary, this study investigated the potential of iPSC-derived MSCs as a promising alternative to traditional MSCs, particularly in cancer research. By comparing the effects of Exos derived from iMSC and WJMSC on cancer cells, we found that the Exos source significantly influenced their biological impact. iMSC-derived Exos demonstrated distinct advantages, particularly in their ability to address challenges associated with traditional MSCs, such as cellular heterogeneity and limited expansion capacity. These findings underscore the importance of further research into iPSC-MSCs to advance cancer therapies and develop improved cell-free therapeutic approaches.

## Figures and Tables

**Figure 1 ijms-26-06178-f001:**
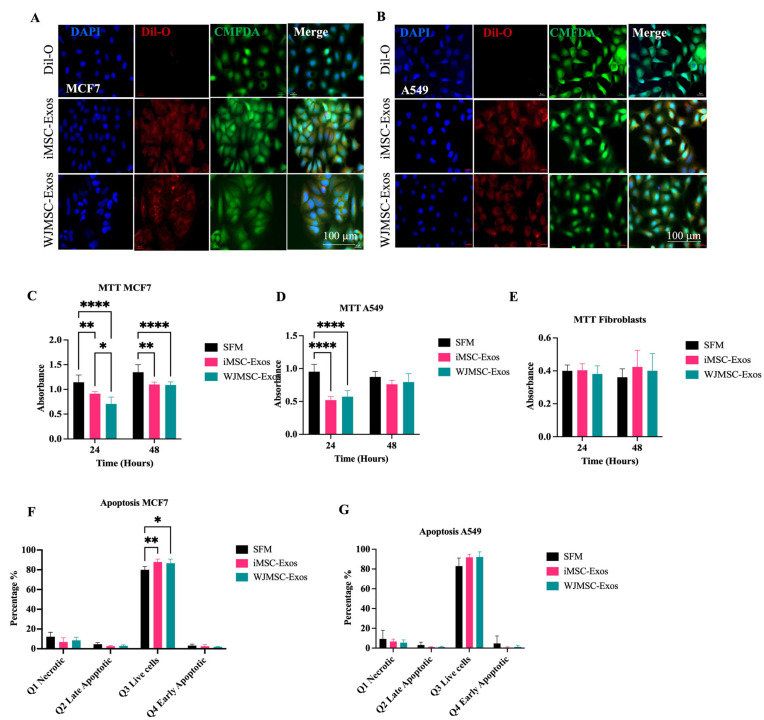
Evaluation of proliferation and apoptosis in MCF7 and A549 cells treated with iMSC-Exos or WJMSC-Exos. (**A**) Internalization of iMSC- and WJMSC-Exos by MCF7 and (**B**) A549 cancer cells. MTT assay in (**C**) MCF7, (**D**) A549 and (**E**) fibroblast cells treated with iMSC- and WJMSC-derived Exos. Statistical analysis of the percentage of Q1 (necrotic cells), Q2 (late apoptotic cells), Q3 (live cells), and Q4 (early apoptotic cells) in (**F**) MCF7 cells and (**G**) A549 cells. Data are represented as mean ± SD. * *p* ≤ 0.05, ** *p* ≤ 0.01, **** *p* ≤ 0.0001.

**Figure 2 ijms-26-06178-f002:**
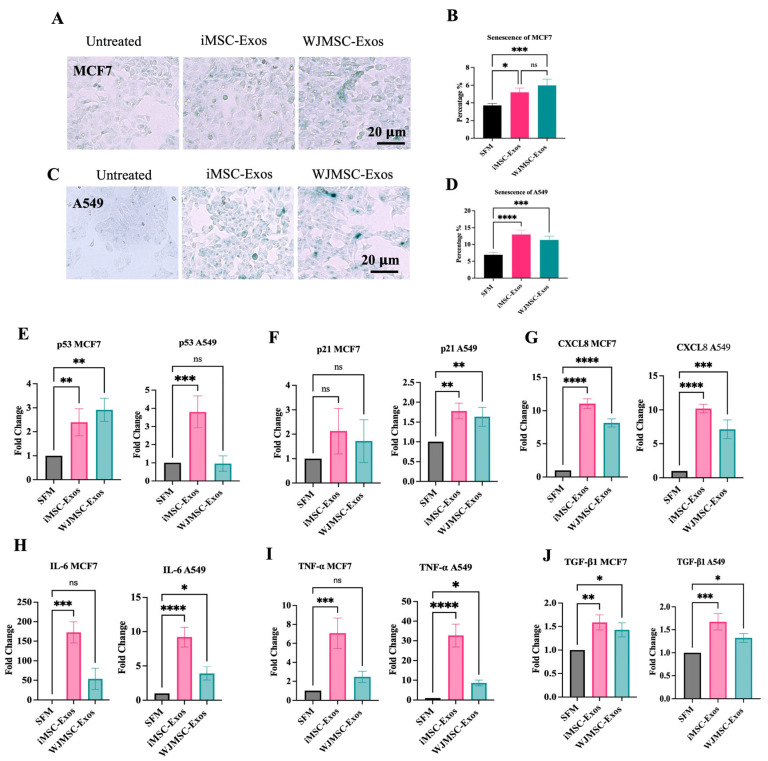
Examination of senescence-associated markers in MCF7 and A549 cells treated with exosomes. ((**A**) & (**C**)) Representative microscopic images of senescent cells (green) in MCF7 (**upper left**) and A549 (**lower left**). ((**B**) & (**D**)) Quantification of senescent cell percentages following treatment with iMSC-derived or WJMSC-derived exosomes for MCF7 cells (**top right**) and A549 cells (**bottom right**) cells. Expression levels of senescence-associated genes in MCF7 and A549 cells treated with iMSC- or WJMSC-derived Exos: (**E**) p53, (**F**) p21^Cip1^, (**G**) CXCL8, (**H**) IL-6, (**I**) TNF-α, and (**J**) TGF-β1. All Data are expressed as mean ± SD. Statistical significance is indicated as ns (not significant), * *p* ≤ 0.05, ** *p* ≤ 0.01, *** *p* ≤ 0.001 and **** *p* ≤ 0.0001.

**Figure 3 ijms-26-06178-f003:**
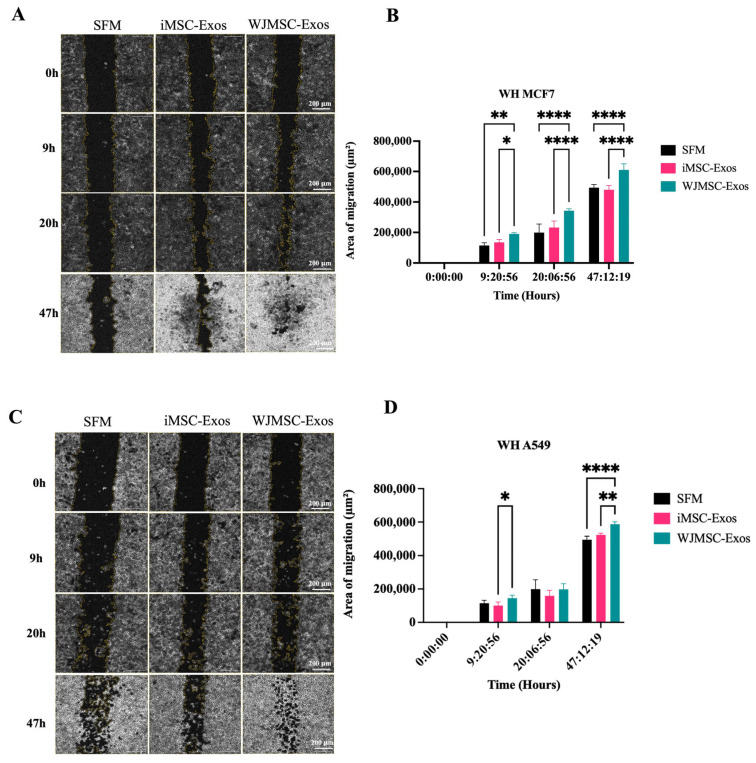
Assessment of migration potential of MCF7 and A549 cells treated with exosomes. Representative microscopic images of wound healing (WH) following a scratch assay are shown for (**A**) MCF7 cells and (**C**) A549 cells. The migration area of (**B**) MCF7 and (**D**) A549 cells treated with either iMSC- or WJMSC-derived exosomes was measured at approximately 9-, 20-, and 47-h post-scratch. Data are presented as mean ± SD. Statistical significance is indicated as * *p* ≤ 0.05, ** *p* ≤ 0.01, and **** *p* ≤ 0.0001.

## Data Availability

Data is contained within the article and Appendix A.

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
