# Peer review of "Exosomes Derived from Induced and Wharton’s Jelly-Derived Mesenchymal Stem Cells Promote Senescence-like Features and Migration in Cancer Cells"

_ijms, 2025, doi:10.3390/ijms26136178_

Round 1

Reviewer 1 Report

Comments and Suggestions for Authors

1. The research objective is not clear enough.

Is the research aiming to explore the feasibility of iMSC-Exos replacing classic MSC-Exos, or to investigate the effect of Exos on tumor proliferation? If it is the former, the focus should be on the consistency of their effects, such as the common effects of MSC-Exos and the consistency of their material basis, including RNA, DNA, lipids, and proteins. If it is the latter, the focus should be on the molecular mechanism of how Exos inhibits proliferation. Simply observing the phenotype of senescence is too superficial and fails to reflect the depth and significance of the research. It is suggested to reconstruct the framework of the paper, for example, focusing on iMSC-Exos to explore their effect on tumor inhibition and the related molecular mechanisms, and finally compare their effects with those of classic MSC-Exos to demonstrate their potential as a replacement.

2. The research indicators are not sufficient enough.

Senescence is a complex biological process, and there is currently no single indicator that can confirm senescence. The use of only SA-β Gal as an indicator to study senescence is not convincing. It is recommended to supplement other indicators commonly used to study senescence, such as telomerase activity and length, P16, P21, and SASP.

3. The discussion part is not closely combined with existing literature.

The discussion part only provides a simple analysis and explanation of the research results. The combination with existing literature is not tight enough to fully reflect the position and contribution of this research in the field. For example, when mentioning the effects of exos on cell proliferation and migration, the differences in the effects of exos from different sources are only briefly compared. The discussion lacks an in-depth exploration of how these findings relate to existing research results on the role of MSC-Exos in cancer, and how they complement, expand, or challenge the existing knowledge. It is suggested to enhance the academic depth and persuasiveness of the paper by increasing the citation and in-depth analysis of relevant literature, comparing the research results with existing ones more comprehensively, explaining how this study corroborates or supplements previous research, and highlighting any new viewpoints or discoveries. Additionally, for unique or unconventional results in this study (such as WJMSC-Exos promoting cancer cell migration while iMSC-Exos do not), further discussion should be conducted in combination with the literature. For instance, explore whether these results are related to the differences in the properties of MSCs from different sources, the composition of exos, and the heterogeneity of cancer cells.

4. The limitations of the study are not mentioned.

The conclusion part of the paper emphasizes the findings and significance of the research but does not address its limitations. For instance, only MCF7 and A549 cancer cell lines are used in the experiments, which may not fully represent the characteristics of all types of breast and lung cancer cells, not to mention other types of cancer. Moreover, the optimization and exploration of experimental conditions such as exos dosage and exposure time may also be insufficient. These limitations could affect the generalizability and extrapolation of the research results. It is suggested to objectively analyze and mention the limitations of the study in the discussion or conclusion part, such as the limitations of the selected cancer cell lines and experimental conditions, and propose directions for future research and improvement suggestions. This will make the paper more complete and rigorous, enabling readers to have a more comprehensive understanding of the actual situation and value of the research.

Author Response

  1. The research objective is not clear enough. Is the research aiming to explore the feasibility of iMSC-Exos replacing classic MSC-Exos, or to investigate the effect of Exos on tumor proliferation? If it is the former, the focus should be on the consistency of their effects, such as the common effects of MSC-Exos and the consistency of their material basis, including RNA, DNA, lipids, and proteins. If it is the latter, the focus should be on the molecular mechanism of how Exos inhibits proliferation. Simply observing the phenotype of senescence is too superficial and fails to reflect the depth and significance of the research. It is suggested to reconstruct the framework of the paper, for example, focusing on iMSC-Exos to explore their effect on tumor inhibition and the related molecular mechanisms, and finally compare their effects with those of classic MSC-Exos to demonstrate their potential as a replacement.

Response: We thank the reviewer for the insightful and constructive feedback. We would like to clarify that the primary focus of our study is to investigate the effects of iMSC-derived exosomes on tumor cells, particularly their potential to suppress tumor proliferation. In response to the reviewer’s comments, we have revised the Introduction to clearly reflect this focus and highlight the novel aspects of our work. While the role of exosomes in cellular senescence has been previously studied in other biological contexts, to the best of our knowledge, the role of iMSC-derived exosomes in inducing senescence specifically in cancer cells has not been thoroughly explored. Previous studies have primarily focused on exosomes as components of the senescence-associated secretory phenotype (SASP) rather than as mediators of senescence induction in cancer cells. Our study aims to fill this important gap by demonstrating a senescence-like response in cancer cells following treatment with iMSC-derived exosomes. Regarding the reviewer’s suggestion to compare iMSC-Exos with classical MSC-Exos, we fully agree that such a comparison would provide further valuable insights. At this stage, it is technically challenging to perform a comprehensive comparison between iMSCs and other types of primary MSC sources due to variability in donor sources and ethical restrictions. Nonetheless, our manuscript includes a comparison between iMSCs and Wharton's Jelly-derived MSCs (WJMSCs), which are commonly used and well-characterized. This provides a valuable basis for future work and opens avenues for more comprehensive comparative studies aimed at assessing the potential of iMSC-Exos as a functional replacement for classical MSC-Exos. To address the reviewer’s concern, we have added a detailed discussion of this limitation in the revised Discussion section, explicitly identifying the absence of a wider MSC-iMSC exosome comparison as a key area for future investigation. We believe these revisions and clarifications strengthen the rationale and impact of our study and more clearly define the novelty and significance of our findings within the context of MSC research.

  1. The research indicators are not sufficient enough. Senescence is a complex biological process, and there is currently no single indicator that can confirm senescence. The use of only SA-β Gal as an indicator to study senescence is not convincing. It is recommended to supplement other indicators commonly used to study senescence, such as telomerase activity and length, P16, P21, and SASP.

Response: We appreciate the reviewer’s important observation regarding the complexity of senescence and the limitations of relying solely on SA-β-Gal staining. In response, we have first addressed this concern by referring to all the findings in this work indicative of senescence as “senescence-like” and “senescence-like features” to more carefully reflect the findings and acknowledge the limitations of the assays used.

We agree with the reviewer that establishing senescence induction would require further marker and functional testing.  Secondly, to strengthen our conclusions, we conducted additional molecular assessment of the expression of several key senescence- and SASP-related genes, including P53, P21, CXCL8, IL-6, TNF-α and TGF-β, among others (Figure 2 in the revised manuscript). These additional results presented in Figure 2 provide further support for the induction of senescence-like phenotype and help to corroborate the SA-β-Gal staining results. Our newly added data further confirmed that both MCF7 and A549 cells exhibited senescence-like responses to exosome treatment. The new data has been incorporated into the revised manuscript.  We also appreciate the reviewer’s suggestion to include telomere-related parameters and expression of p16 as additional markers of senescence. However, we would like to clarify that telomere dysfunction is not a reliable or commonly used indicator of senescence in tumor cells. This is largely because most cancer cells have activated telomerase or alternative telomere maintenance mechanisms, allowing them to bypass telomere shortening. Thus, it was long shown that tumor cell senescence is most likely to be telomere length-independent (PMID: 12101184). As a result, telomere length does not consistently reflect senescence status in tumor cell models as it would in replicative senescence models. Similarly, while p16 is a classical senescence marker in primary cells, it is frequently mutated, deleted, or epigenetically silenced in many tumor cell lines including those used in this work (PMID: 10632371). This makes it an unreliable indicator of senescence in cancer models, as its expression may not correlate with the actual senescence phenotype (PMID: 32235364). Subsequently, we opted to use alternative, more reliable markers for our cancer cell models. We hope the inclusion of new data to support the induction of a senescence-like state in response to iMSC derived exosomes and this explanation adequately address to the reviewer’s concern on this issue and enhances the rigor and interpretability of our findings.

  1. The discussion part is not closely combined with existing literature. The discussion part only provides a simple analysis and explanation of the research results. The combination with existing literature is not tight enough to fully reflect the position and contribution of this research in the field. For example, when mentioning the effects of exos on cell proliferation and migration, the differences in the effects of exos from different sources are only briefly compared. The discussion lacks an in-depth exploration of how these findings relate to existing research results on the role of MSC-Exos in cancer, and how they complement, expand, or challenge the existing knowledge. It is suggested to enhance the academic depth and persuasiveness of the paper by increasing the citation and in-depth analysis of relevant literature, comparing the research results with existing ones more comprehensively, explaining how this study corroborates or supplements previous research, and highlighting any new viewpoints or discoveries. Additionally, for unique or unconventional results in this study (such as WJMSC-Exos promoting cancer cell migration while iMSC-Exos do not), further discussion should be conducted in combination with the literature. For instance, explore whether these results are related to the differences in the properties of MSCs from different sources, the composition of exos, and the heterogeneity of cancer cells.

Response: We thank the reviewer for this constructive feedback. We fully agree that our original discussion required a stronger connection with the existing body of literature. In response, we have significantly revised and expanded the Discussion section to include more thorough discussion. Specifically, we have incorporated relevant literature demonstrating the complex and sometimes contradictory roles of MSC-derived exosomes in cancer, which are influenced by factors such as cell source, culture conditions, and the type of tumor model used. While both WJMSC-Exos and iMSC-Exos showed comparable anti-proliferative and pro-senescent effects in our cancer models, only WJMSC-Exos enhanced migration, contributing to the ongoing debate on the divergent effects of MSC-Exos in cancer. We also added a note in the Limitations and Future Perspectives section at the end of the discussion emphasizing the need for comprehensive cargo analysis to understand the molecular basis of these functional differences, as this aspect was not covered in our current study. These additions have been incorporated throughout the revised Discussion to provide a clearer and more rigorous interpretation of our results.

  1. The limitations of the study are not mentioned. The conclusion part of the paper emphasizes the findings and significance of the research but does not address its limitations. For instance, only MCF7 and A549 cancer cell lines are used in the experiments, which may not fully represent the characteristics of all types of breast and lung cancer cells, not to mention other types of cancer. Moreover, the optimization and exploration of experimental conditions such as exos dosage and exposure time may also be insufficient. These limitations could affect the generalizability and extrapolation of the research results. It is suggested to objectively analyze and mention the limitations of the study in the discussion or conclusion part, such as the limitations of the selected cancer cell lines and experimental conditions, and propose directions for future research and improvement suggestions. This will make the paper more complete and rigorous, enabling readers to have a more comprehensive understanding of the actual situation and value of the research.

Response: We thank the reviewer for their valuable feedback. We are in full agreement that the use of only two cancer cell lines—MCF7 (breast cancer) and A549 (lung cancer)—does not capture the full heterogeneity of breast and lung cancers, nor the broader spectrum of other tumor types. However, both cell lines remain widely used in cancer research as well-established in vitro models. To resolve this point, we have now explicitly included this important point in the revised Discussion section, stating that further studies should assess a wider panel of cancer and non-cancerous cell lines with varying molecular subtypes to validate the generalizability in our findings. This limitation is now clearly acknowledged. Additionally, we recognize that the experimental conditions, such as exosome dosage and exposure duration, were based on preliminary optimization and may not reflect the full therapeutic window. Accordingly, we have updated the Discussion section and highlighted the need for systematic dose–response and time-course studies in future work. By explicitly acknowledging these limitations, we aim to provide a more rigorous and transparent interpretation of our findings and to guide future research toward enhancing the translational potential of iMSC-derived exosomes in cancer therapy. We have also incorporated some highly relevant suggestions for future studies as suggested by the reviewer. We hope that this comprehensive revision of the Discussion in response to comments 3 and 4 address the reviewer’s concerns satisfactorily.

Note to the Reviewer
We would like to inform the reviewer that all MSC and exosome characterization data have been moved to the Supplementary File, as no significant differences were observed between exosomes derived from the two sources. We believe that presenting this information in the supplementary section helps streamline the main manuscript and maintain its focus on experiments directly related to cancer cell phenotypes and the disease mechanisms.

Reviewer 2 Report

Comments and Suggestions for Authors

Author Response

  1. Please could you justify the use of Wharton Jelly MSCs as a source of your exos and not any other source that you have listed in lines 44-45?

Response: Thank you for your comment. We have now clarified in the Introduction that WJMSCs were selected due to their high proliferation capacity, immunomodulatory properties, and ease of isolation (Page 2, Lines 82-92), which make them a widely used and well-characterized MSC source in regenerative medicine and cancer research.

  1. You have justified the use of induced MSC-Exos, however – induced MSCs exos are different to normal MSC-exos and could potentially impact the outcome of results. Please could you acknowledge this in your methods and in your discussion. There is plenty literature available on the topic.

Response: Thank you for the valuable comment. We have now acknowledged in the Discussion that although direct comparative studies between iMSC- and MSC-derived exosomes remain limited, existing evidence highlights differences in their molecular composition and functional properties. Additionally, accumulating research emphasizes the promising therapeutic potential of iMSC-derived exosomes (Pages 8-9, Lines 235-243), particularly due to their scalability and reduced donor variability.

  1. Please could you define SFM when it is first mentioned in the results. What does it stand for and what is it that makes it an ideal control? Currently it is unclear what is your control. Kindly clarify in text.

Response: Thank you for pointing this out. We have now defined SFM (serum-free medium) at its first mention in the Results section for clarity. We also clarified that SFM was used as a negative control because it lacks fetal bovine serum (FBS), which is known to include extracellular vesicles including exosomes. This ensures that the any observed cellular effects in the treatment groups are attributable solely to the MSC-derived exosomes, rather than background vesicular contamination from serum.

  1. Very interesting data – considering that ideally we want cancer cells to adapt to self-destruction programmes; senescence of cancer cells is most likely not a bad approach. However, you will need to further examine this in non-cancer and healthy cells to confirm if MSCs-exos really have a senescent effect in health (healthy cells). Please add another paragraph in your discussion for ‘limitations and future perspectives’ and add this point.

Response:  We would like to thank the reviewer for this insightful and valuable comment. We agree that further investigation is needed to determine whether the observed senescent-like effects induced by MSC-derived exosomes are specific to cancer cells or if they also affect healthy, non-cancerous cells. This important point has been acknowledged and addressed in the revised discussion as one of the limitations.

  1. Another point to consider is that, if these cells are so senescent, how are they migrating so well in comparison to the control? While you have provided references in your discussion (48-51); are you able to provide a potential mechanism connecting the negative correlation of proliferation and migration; as well as, the correlation between senescence and migration using a using a simple flow chart? This is to add clarity for your readers; as to how these might function differently in different types of cancers. Additionally, are there any recent references (last 2-3 years) for references 48-50?

Response: Thank you for the insightful comment. In response, we have clarified the potential link between senescence and increased migration by expanding the discussion on the senescence-associated secretory phenotype (SASP), which can promote tumor invasiveness and epithelial-mesenchymal transition (EMT). We have included recent supporting references to strengthen this point. Additionally, to aid clarity, we have added a simplified flow chart illustrating the potential mechanisms connecting senescence, and increased migration, as well as how these effects may vary across different cancer types (Supplementary figure 2)

Figure S2. . Schematic summary of exosome-induced senescence and migration effects in cancer cells. iMSC- and WJMSC-derived exosomes induced senescence in MCF7 and A549 cancer cells. However, only WJMSC-Exos promoted migration through a senescence-like state in these cancer cells, while iMSC-Exos did not affect migration, highlighting a key functional difference between the two exosome types in modulating cancer cell behavior.

  1. In figure 6 – migration images, please could you add the scale?

Response: Thank you for pointing this out. We have now added scale bars (200 µm) to the migration images in Figure 6.

  1. Any particular reason you chose 47hours as time point? It is usually 48 hours. Response: Thank you for the observation. We agree that 48 hours is the more commonly reported time point; however, to remain accurate, we reported the actual time at which imaging was conducted (47 hours and 12 minutes), rounding it to 47 hours in the text. That said, if the reviewer prefers it to be presented as the standard 48-hour time point, we would be happy to adjust it accordingly.
  2. In future work – please also add gene expression of the senescent cells and ELISA of the media of the senescent cells – these experiments will help determine if SASP is actually present in these senescent cells and you must compare these with healthy cells, as well as with other cancer cell lines like HeLa etc.

Response: We would like to thank the reviewer for pointing this out. In the revised Discussion, we have now included future directions that involve a detailed assessment of SASP involvement in the decreased proliferation and increased in migration by evaluating gene expression and performing ELISA on the media of senescent cells. We also acknoledged the importance of including additional cancer cell lines as well as healthy non-cancerous cells for comparison to improve the relevance of the findings.

  1. Overall – while your work is well presented, please acknowledge all your limitations and add that in your future work as outlines above. All the very best.

Response: Thank you for your positive words and helpful suggestions. We have now acknowledged the study’s limitations in the revised Discussion, including the use of only two cancer cell lines, the lack of in vivo models, and the limited optimization of experimental conditions such as exosome dosage and exposure duration (Page 10-11, Lines 343-357). We have emphasized these as important areas for future research to enhance the translational relevance of our findings.
